# Polymorphism rs2383207 of CDKN2B-AS and Susceptibility to Atherosclerosis: A Mini Review

**DOI:** 10.3390/ncrna8060078

**Published:** 2022-11-18

**Authors:** Sofia Vladimorovna Timofeeva, Tatiana Alexandrovna Sherchkova, Tatiana Pavlovna Shkurat

**Affiliations:** The Department of Genetics, Southern Federal University, 34006 Rostov-on-Don, Russia

**Keywords:** CDKN2B antisense (CDKN2B-AS), lncRNA, gene polymorphisms, atherosclerosis, meta-analysis

## Abstract

We conducted this meta-analysis to estimate associations between CDKN2B antisense (CDKN2B-AS) rs2383207 polymorphism and susceptibility to atherosclerosis. A systematic literature research of Google Scholar and PubMed was performed to identify eligible studies. Overall, eight studies were included for meta-analyses. The association was assessed by statistical odds’ ratio (OR) with 95% confidence interval (CI). RevMan software (Cochrane Collaboration, 5.3. Copenhagen) was used for the meta-analysis. Pooled overall analyses showed that rs2383207 polymorphism was associated with the risk of atherosclerosis in the whole population. Additional analyses by ethnicity revealed that rs2383207 polymorphism was associated with susceptibility to atherosclerosis in Asians and Caucasians. Our results suggest that rs2383207, might serve as genetic biomarkers of atherosclerosis. Further, studies will be required to confirm the observed association.

## 1. Introduction

Cardiovascular disease (CVD) is one of the main causes of death in the world. According to data provided by the WHO, 17.9 million people die each year from CVDs, which is about 32% of all deaths in the world. Contributions to the pathogenesis of CVD include genetic and non-genetic factors such as environmental, behavioural and others. Number of premature deaths can be reduced by reducing behavioural risk factors such as smoking, drinking of alcohol, obesity and others.

Atherosclerosis underlies cardiovascular diseases and remains a leading cause of morbidity and mortality worldwide. The molecular mechanisms involved in the development of atherosclerosis are not fully understood. Noncoding RNAs (ncRNAs) are transcribed but not translated into proteins, performing their biological functions at the RNA level [1]. Accruing evidence has disclosed ncRNAs regulate pivotal cellular and molecular processes during all stages of atherosclerosis including cell growth, invasion and survival, expression and release of pro- and anti-inflammatory intermediaries, cellular uptake and efflux of lipids, and proteolytic balance [2]. CDKN2B antisense ncRNA (CDKN2B-AS) is located on human chromosome 9p21, a region that has been repeatedly related with atherosclerosis [3]. While the exact role of CDKN2B-AS is still uncertain, it has been shown that the expression levels of several neighbour protein-coding genes such as cyclin-dependent kinase inhibitors 2A (CDKN2A), CDKN2B, and methylthioadenosine phosphorylase (MTAP) are regulated by CDKN2B-AS [4]. Previous studies have demonstrated that overexpression of CDKN2B-AS1 hinders vascular smooth muscle cells (VSMCs) proliferation and accelerates apoptosis via inhibiting the PI3K/AKT pathway, and also that CDKN2B-AS may promote atherosclerosis by impacting thrombogenesis, and plaque stability [4,5]. Recently, studies have already investigated potential associations between several variants CDKN2B-AS polymorphisms and the susceptibility to atherosclerosis, with conflicting results. Thus, we chose rs2383207 polymorphism and performed the present meta-analysis to obtain a more conclusive result.

## 2. Results

We have found 122 potentially relevant papers. Among these articles, eight eligible studies were included for pooled analyses (Figure 1). The NOS score of eligible articles ranged from seven to eight, which indicated that they were of high quality. Baseline characteristics of included studies are shown in Table 1.

### 2.1. Characteristics of Included Studies

In this meta-analysis, 6894 participants were included: 3277 in the group with atherosclerosis and 3617 in the control group. Eight articles were selected: three for rs2383207 in the Caucasian population [6,7,8]; five for rs2383207 in the Asian population [9,10,11,12,13].

Overall allele frequencies were obtained by pooling the alleles of each genotype in focus groups. All investigated polymorphisms contain a major (M) and a minor (m) alleles. So the allele distributions in investigated polymorphisms denominated as MM/Mm/mm.

### 2.2. Overall and Subgroup Analyses

To investigate potential associations of CDKN2B-AS rs2383207 polymorphism with susceptibility to atherosclerosis eight studies were included for analyses. Significant association with atherosclerosis was observed for rs2383207 only in heterozygous model (*p* = 0.007, OR 0.86, 95%CI: 0.78–0.96). No significant associations were observed in overall analyses for other genetic models (allele model: *p* = 0.20, OR 0.91, 95%CI: 0.79, 1.05; recessive model: *p* = 0.47, OR 0.92, 95%CI: 0.73–1.16; dominant model: *p* = 0.13, OR 0.78, 95%CI: 0.71–0.86; homozygous model: *p* = 0.19, OR 0.80, 95%CI: 0.57–1.12).

Further, subgroup analyses by ethnicity revealed that rs2383207 investigated polymorphism were significantly associated with susceptibility to atherosclerosis in Asians (homozygous model: *p* = 0.002, OR 0.76, 95%CI: 0.61–0.91). Moreover, rs2383207 polymorphism was associated with susceptibility to atherosclerosis in Caucasians (allele model: *p* = 0.16, OR 1.12, 95%CI: 0.96–1.30; recessive model: *p* = 0.92, OR 1.04, 95%CI: 0.47–2.29), Table 2.

### 2.3. Sensitivity Analyses

We accomplished sensitivity analyses to test stabilities of pooled results by excluding studies that violated the HWE. Altered results were not noticed in any comparisons, which suggested that our findings were statistically stable.

### 2.4. Publication Biases

We used funnel plots to assess publication bias. We did not find obvious asymmetry of funnel plots in any comparisons, which suggested that our findings were unlikely to be severely impacted by publication bias.

## 3. Discussion

This study has evaluated the association of genetic polymorphism rs2383207 (9p21.3) with occurrence of atherosclerosis in Caucasian and Asian populations. Previous studies reported about the association of rs2383207 (A) risk allele with cardiovascular diseases (CAD) [14], presentation of CAD [15] and severity of CAD [12]. Akinyemi et al. reported that rs2383207 increases ischemic stroke (IS) incidence in indigenous West African men [16]. Furthermore, GWAS was also used to demonstrate that the CDKN2B-AS variant rs2383207 increases the risk of IS and coronary heart disease in Caucasian populations [17,18]. Notably, studied chromosome 9p21 variants in Chinese populations, they concluded that mutations in rs2383207 may reduce the risk of IS [14]. Additionally, Zhou et al. demonstrated that the SNP rs2383207 on chromosome 9p21 is significantly associated with CHD in Chinese Han population. The risk allele of the SNP rs2383207 plus family history of CHD have a cumulative, significant association with CHD [9]. In the present study, we focused on the relationship between atherosclerosis and rs2383207.

The relationship between CDKN2B-AS and atherosclerosis has been studied intensively. However, no consistent conclusion has been reached. In this meta-analysis, 6894 participants were included; the atherosclerosis and control groups contained 3277 and 3617 individuals, respectively. Statistical significance existed only in heterozygous model comparisons of rs2383207. With the significant outcome in allele comparison, we can come to the conclusion that carrying the mutated allele G on rs2383207 may be associated with atherosclerosis. Subgroup analysis represented potential association between rs2383207 and atherosclerosis for recessive model in the Caucasian subgroup and for homozygous model in the Asian population. However, studies have not included information about influence of environmental factors. Hence, there was no additional analysis of environmental factors such as smoking and drinking alcohol, which may also be associated with atherosclerosis risk.

In this mini-review, a potential association between rs2383207 polymorphism in CDKN2B-AS locus and susceptibility to atherosclerosis was analysed. As a result, we have supposed that rs2383207 may be a potential marker for atherosclerosis in Caucasian and Asian populations. This result is quite different from those of previous meta-analyses. Further large scale population based study is needed to consolidate our findings. Additionally, we acknowledge the important role of conducting genome wide association study in identifying the loci that are associated with CVDs. So, research on the influence of CDKN2B-AS with other potential genetics loci, and the environmental factors on CVDs pathogenesis need to be considered with large, representative samples.

## 4. Materials and Methods

### 4.1. Literature Search and Inclusion Criteria

This meta-analysis conformed with the Preferred Reporting Items for Systematic Reviews and Meta-analyses (PRISMA) guideline [19]. Potentially related studies published prior to September 2022 were extracted from Google Scholar and PubMed using the following key phrases: (CDKN2B-AS long non-coding RNA OR CDKN2B antisense RNA) and (allele OR mutation OR genotype OR variant OR polymorphism) and (atherosclerosis OR coronary heart disease).

The inclusion criteria for publications in this meta-analysis were: (1) studies evaluating the association between the rs2383207 polymorphism and the risk of atherosclerosis using the case–control method; (2) diagnosis of atherosclerosis confirmed by ultrasound evaluation of the carotid arteries; (3) studies with or without Hardy–Weinberg equilibrium (HWE) bias for the control group; (4) DNA copy number was measured using previously described methods based on PCR technology; (5) the sample only included adults over the age of 18; and (6) all data were presented as OR and 95% CI.

The main exclusion criteria were: (1) reviews, short communications, conference abstracts, comments; (2) studies without case–control statistics; (3) studies without providing sample data; (4) research in the field of cerebrovascular diseases; and (5) duplicate data.

### 4.2. Data Extraction and Quality Assessment

In accordance with the selection criteria, all publications were independently reviewed by two reviewers. The extracted data included basic information: (a) first author name, (b) year of publication, (c) country of origin, (d) ethnicity, (e) genotyping method, (f) sample size, (g) allele and genotype frequencies rs2383207. In addition, we also assessed the methodological quality of the publications included in our meta-analysis using the Newcastle–Ottawa scale (NOS) [20]. The NOS contains 8 items in 3 areas, and the total maximum score is 9 [21].

### 4.3. Statistical Analyses

The data were analysed using Review Manager 5.3 software (RevMan 5.3) to calculate odds ratio (OR) with 95% confidence interval (CI). Heterogeneity was assessed in all studies using the Cochran and I squared tests [22,23]. The total OR was calculated for the dominant (GG + CG vs.CC), recessive (GG vs. CC + CG), and allelic (C vs. G) models, as well as for the homozygous (GG vs. CC) and heterozygous (CG vs. CC) models. Z-scores were used to assess the statistical significance of pooled ORs, and a *p*-value < 0.05 was considered statistically significant. I squared statistics were used to assess interstudy heterogeneity. If I squared was higher than 50%, random effects models were used to pool the data. Otherwise, fixed-effect models were chosen. Subgroup analyses by ethnicity of participants and type of disease were also performed. Stabilities of synthetic results were evaluated with sensitivity analyses, and publication bias was evaluated with funnel plots.

## 5. Conclusions

Thus, based on the obtained results, it can be assumed that the CDKN2B-AS rs2383207 polymorphism may be associated with the risk of atherosclerosis among the Asian and Caucasian populations. More studies with larger samples from other ethnic groups are needed to refute or confirm the current findings.

## Figures and Tables

**Figure 1 ncrna-08-00078-f001:**
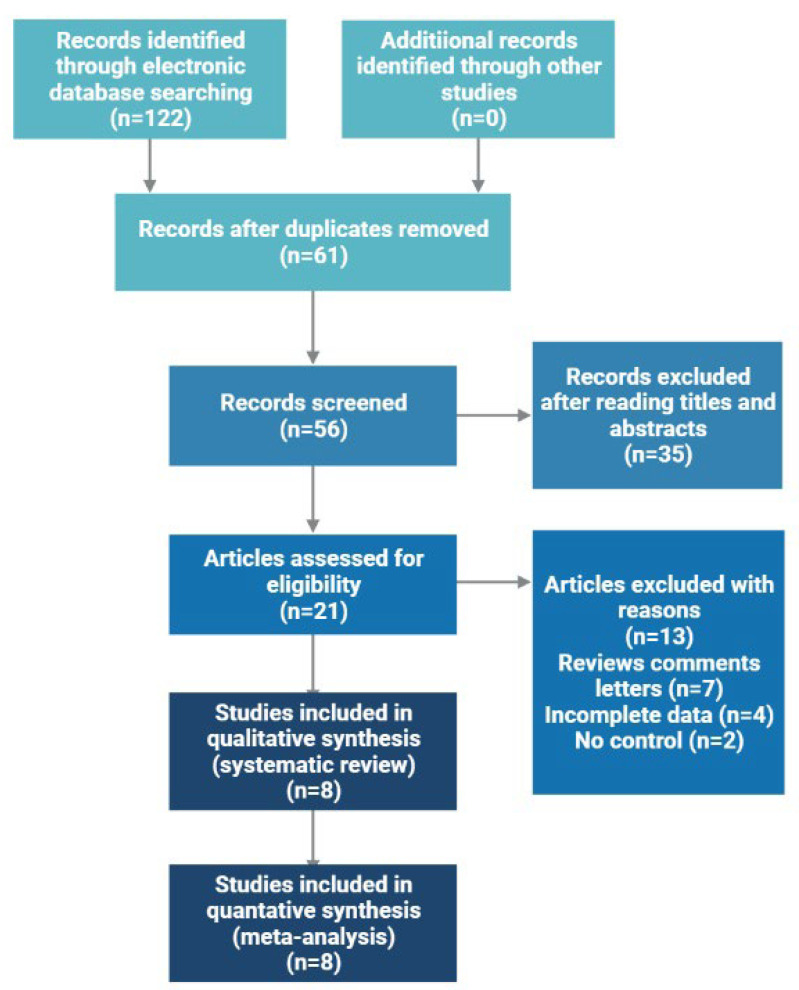
Flowchart of study selection for the present study.

**Table 1 ncrna-08-00078-t001:** The characteristics of included studies.

First Author, Year	Country	Ethnicity	Sample Size	Genotype Distribution (MM/Mm/mm)		*p* Value for HWE	NOS Score
				Cases Control			
Abdulah, 2008	United States	Caucasian	310/560	139/121/50	147/277/136	0.88	8
Zhou, 2008	China	Asia	1360/1360	702/520/138	592/605/163	0.75	8
Chen, 2009	China	Asia	212/232	107/69/36	71/114/47	0.96	8
Kumar, 2011	India	Asia	301/424	137/124/40	174/190/60	0.63	7
Cakmak, 2015	Turkey	Caucasian	220/240	83/101/36	102/118/20	0.21	7
El-Menyar, 2015	Egypt	Asia	236/152	146/77/12	84/58/10	1	7
Yang, 2018	China	Asia	540/548	247/236/57	244/251/53	0.49	8
García-González, 2021	Yucatan	Caucasian	98/101	23/53/22	26/51/24	0.94	8

Abbreviations: HWE, Hardy–Weinberg equilibrium; NOS, Newcastle–Ottawa scale.

**Table 2 ncrna-08-00078-t002:** Results of overall and subgroup analyses.

Genetic Model	Population	Number of Studies	Test of Association		Test of Heterogeneity		
			OR (95%CI)	*p*	*P* _h_	I^2^, %	Model
C vs. G	Total	8	0.91 [0.79, 1.05]	0.20	0.002	70	R
	Asian	5	0.82 [0.71, 0.95]	0.007	0.05	57	R
	Caucasian	3	**1.12 [0.96, 1.30]**	0.16	0.33	9	F
CG vs. CC	Total	8	**0.86 [0.78, 0.96]**	**0.007**	0.0009	72	R
	Asian	5	0.85 [0.75, 0.95]	0.005	0.001	65	R
	Caucasian	3	0.91 [0.70, 1.19]	0.48	0.02	85	R
GG vs. CC	Total	8	0.80 [0.57, 1.12]	0.19	0.0002	75	R
	Asian	5	**0.76 [0.61, 0.91]**	**0.002**	0.26	24	F
	Caucasian	3	0.94 [0.30, 2.97]	0.92	<0.0001	91	R
GG + CGvs.CC	Total	8	0.78 [0.71, 0.86]	0.13	<0.00001	37	F
	Asian	5	0.79 [0.70, 0.88]	0.12	<0.0001	46	F
	Caucasian	3	0.78 [0.63, 0.96]	0.15	0.02	47	F
GGvs.CC + CG	Total	8	0.92 [0.73, 1.16]	0.47	0.03	54	R
	Asian	5	0.88 [0.75, 1.05]	0.15	0.77	0	F
	Caucasian	3	**1.04 [0.47, 2.29]**	0.92	0.001	85	R

Abbreviations: CI, confidence interval; OR, odds ratio; F, fixed model; R, random model.

## Data Availability

Data about human genotyping was performed in table with links to study. We didn’t conduct own genotyping research with patients.

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
