# Peer review of "Polymorphism rs2383207 of CDKN2B-AS and Susceptibility to Atherosclerosis: A Mini Review"

_ncrna, 2022, doi:10.3390/ncrna8060078_

Round 1

Reviewer 1 Report

the study by Timofeeva et al analyzes the potential association of the (CDKN2B-AS) rs2383207 polymorphism and susceptibility to atherosclerosis. They employed the statistical odds ratio (OR)  with 95% confidence interval (CI) and RevMan software to study the association. Study concluded that the rs2383207 is associated  with atherosclerosis in Asians 14 and Caucasions and  populations, and this SNP can be used as a genetic marker for Atherosclerosis. 

The study is fine but I have the following concerns.

-        The title should be Polymorphism rs2383207 of CDKN2B-AS and susceptibility to  atherosclerosis: A mini review

In the introduction the authors should write something about the cardiovascular disease are caused by genetic and non-genetic factor such as life style, smoking, obesity, etc..

-       The authors should acknowledge the role of genome wide association study in identifying the loci that are associated with CVD.

- the authors included few population from Asia and Europe. Therefore, I think it is not easy to draw the conclusion that this SNP is associated with  atherosclerosis in these 2 populations.

- the recommendation should be large scale future population based study should be conduced. 

-The paper by Zhou et al., Associations between single nucleotide polymorphisms on chromosome 9p21 and risk of coronary heart disease in Chinese Han population.  Arterioscler Thromb Vasc Biol, 2008 Nov;28(11):2085-9. doi: 10.1161/ATVBAHA.108.176065. Epub 2008 Aug 28. Should be included and discussed

Author Response

Dear Reviewer, thank you so much for your time, comments and suggestions. We took into account every comment. We corrected the manuscript for all six comments. 

Reviewer 2 Report

This manuscript by Timofeeva and colleagues presents a comprehensive meta-analysis of association between rs2383207 polymorphism in CDKN2B-AS locus and susceptibility to atherosclerosis. While, the analysis is well performed and presented, authors need to improve the overall readability of the manuscript.

1.     Authors need to revise the manuscript for language, as it suffers from several grammatical and typographical errors.

2.     In keywords, change ‘atherosclerosclerosis’ to ‘atherosclerosis’ and ‘metaanalysis’ to ‘meta-analysis’.

3.     Page #2, line#86, please change’ which tree investigated’ to ‘three investigated’.

Author Response

Dear Reviewer, thank you so much for your time, comments and suggestions. We took into account every comment. We corrected the manuscript for all three comments. 

Round 2

Reviewer 2 Report

No further comments